# *Lelliottia amnigena* and *Pseudomonas putida* Coinfection Associated with a Critical SARS-CoV-2 Infection: A Case Report

**DOI:** 10.3390/microorganisms11092143

**Published:** 2023-08-24

**Authors:** Victoria Birlutiu, Rares-Mircea Birlutiu, Elena Simona Dobritoiu

**Affiliations:** 1Faculty of Medicine, Lucian Blaga University of Sibiu, Str. Lucian Blaga, Nr. 2A, 550169 Sibiu, Romania; 2County Clinical Emergency Hospital, 550245 Sibiu, Romania; 3Clinical Hospital of Orthopedics, Traumatology and Osteoarticular TB Bucharest, B-dul Ferdinand 35-37, Sector 2, 021382 Bucharest, Romania

**Keywords:** *Lelliottia amnigena*, *Pseudomonas putida*, case report, infection, SARS-CoV-2

## Abstract

*Lelliottia amnigena* is a Gram-negative facultative anaerobic bacillus identified from water sources and later from food (onions, cream, unpasteurized milk, and Spanish pork sausages), which, under certain circumstances, can cause infections in humans, especially in immunocompromised patients. Few cases of human infections have been reported in the literature, such as endophthalmitis, urinary tract infection, pyonephrosis, and sepsis. We describe the case of a 69-year-old Caucasian male patient who lives in an urban environment and presents himself to the emergency department with chills, fever, myalgias, marked physical asthenia, dry cough, dyspnea, symptoms for which he is tested and confirmed with SARS-CoV-2 infection using real-time reverse transcriptase–polymerase chain reaction (RT-PCR) from nasal and pharyngeal swabs, after being admitted the same day (25 May 2023) to the Infectious Diseases Clinic from the County Clinical Emergency Hospital Sibiu, Romania. At the time of admission, a pulmonary computerized tomography (CT) scan was performed, which revealed a severity score of 10 out of 25. In the second week of the disease, the patient presents with hemoptysis, from which bacteriological examinations are carried out, and *Pseudomonas putida* and *Lelliottia amnigena* are identified. The evolution was slowly favorable under antiviral treatment, corticotherapy, antibiotic therapy (in the absence of the identified etiology, initially meropenem was administered in association with linezolid, and then ceftazidime-avibactam), voriconazole, anakinra, salbutamol inhaler, inhalation corticosteroids, with slow reduction in oxygen requirement, the patient continued oxygen therapy at home after discharge with a flow rate of 5 L/minute. During the third harvesting of sputum samples, *P. putida* was isolated along with *L. amnigena*, both strains of low-virulence species, and maintained susceptibility to antibiotics. In the context of an immunosuppressed patient with previous pulmonary surgery for actinomycosis, chronic obstructive pulmonary disease, and bronchiectasis, all these conditions are favorable for biofilm formation. *L. amnigena* remains a pathogen rarely isolated in human pathology, but we should pay more attention, especially in the immunosuppressed patient, where it can be responsible for an extremely serious clinical picture.

## 1. Introduction

*Lelliottia amnigena*, originally named *Enterobacter amnigenus* in 1981 [1], subsequently from 2013 known as *L. amnigena* after Brady et al. [2], is a Gram-negative, facultative anaerobic bacillus identified from water sources and later from food (onions, cream, unpasteurized milk, and Spanish pork sausages) [3], which, under certain circumstances, can cause infections in humans. Few cases of human infections have been reported in the literature, such as endophthalmitis [4], urinary tract infection [5], pyonephrosis [6], and sepsis [7], which occur particularly in immunocompromised patients [5].

Identifying new cases can provide additional information on the real incidence of bacteria with low pathogenicity and provide data on their antibiotic susceptibility test results, with different hypotheses being issued regarding their resistance to beta-lactams.

## 2. Case Report

We describe the case of a 69-year-old Caucasian male patient who lives in an urban environment and presents himself to the emergency department with chills, fever, myalgias, marked physical asthenia, dry cough, dyspnea, symptoms for which he is tested and confirmed with SARS-CoV-2 infection using real-time reverse transcriptase–polymerase chain reaction (RT-PCR) from nasal and pharyngeal swabs, after being admitted the same day (25 May 2023) to the Infectious Diseases Clinic from the County Clinical Emergency Hospital Sibiu, Romania. Based on the past medical history of the patient, we report a case of type 2 diabetes mellites under treatment with oral antidiabetic agents, high blood pressure, ischemic heart disease, chronic obstructive pulmonary disease, right pulmonary upper lobe-operated actinomycosis, hepatic steatosis, operated pyloric stenosis, recent right facial paresis, and otitis externa with *Candida parapsilosis.* In terms of COVID-19 immunizations, the patient was vaccinated with three doses of a messenger RNA vaccine.

At the time of admission, a pulmonary computerized tomography (CT) scan was performed, which revealed multiple areas of condensation in the matt glass aspect disseminated bilaterally with fine left posterobasal interlobular septal thickening. A severity score of 10 out of 25 was reported; the lesions involved 5–25% of the right superior lobe, between 5 and 25% of the left middle lobe, 5 and 25% of the right lower lobe, 5 and 25% of the left superior lobe, and 5 and 25% of the left lower lobe (Figure 1). The severity score proposed by Francone M et al. was used to assess lung impairment [8]. Focal changes in centrilobular, panlobular, and paraseptal bilateral superior lobe emphysema, bilateral infrahilar bronchial ectasia, hiatal hernia of 33 × 28 mm, reduced stomach body size, more likely postoperative, with the appearance of a gastro-enteric anastomosis, and arranged paramedian left epigastric are also reported.

The patient presented with acute respiratory failure, which required oxygen therapy with a simple face mask with a flow rate of 5 L/minute and peripheral oxygen saturation of 96%. The patient required high-flow nasal oxygen therapy (with a flow rate of 60 L/minute) to manage the acute hypoxemic respiratory failure exacerbation during the first night of admission. On the third day of admission, a pulmonary CT scan re-evaluation was performed, which confirmed the aggravation of lung impairment (a severity score of 14 out of 25 was reported), which was also maintained at the last assessment on the fourteenth day of hospitalization (Figure 2).

During the first week of hospitalization, two bacteriological examinations of the specimens obtained from the lower respiratory tract (sputum) were performed, but all cultures were negative.

In the second week of the disease (day seven), the patient presented with hemoptysis. After this episode, another bacteriological examination of the specimens obtained from the lower respiratory tract (sputum) was performed. An assessment of the sputum quality was performed as a routine to estimate the amount of oropharyngeal contamination, an examination that is performed via microscopic examination of the cellular components in a stained smear under low power field magnification. A reduced number of squamous epithelial cells and the presence of more than 25 leukocytes per low-power field suggested that the samples were derived from the site of an active infection. Also, previous examinations of the two samples collected during the first week of hospitalization were harvested in the same process after the patient was instructed on how to collect the sputum. All samples were collected in the morning under the supervision of a nurse, also the patient was not using antiseptic mouthwash and did not have a meal prior to sampling. From the sputum cultures that were carried out (2 cultures from 2 samples), *Pseudomonas putida* and *Lelliottia amnigena* were identified from both samples after three days. The isolated bacteria were identified using a VITEK 2 Compact analyzer (bioMérieux, Marcy-l’Étoile, France). Minimum inhibitory concentrations were assessed according to the Clinical and Laboratory Standards Institute (CLSI) MIC breakpoints (CLSI M100 32nd edition) [9].

The *Pseudomonas putida* strain was sensitive to amikacin, cefepime, ceftazidime, ciprofloxacin, gentamicin, imipenem, pefloxacin, piperacillin, piperacillin/tazobactam, and tobramycin, and resistant to ticarcillin, ticarcillin/clavulanic acid, and trimethoprim/sulfamethoxazole. *Lelliottia amnigena* strain was sensitive to amikacin, cefepime, ceftazidime, ceftriaxone, ciprofloxacin, gentamicin, piperacillin, tobramycin, trimethoprim/sulfamethoxazole, levofloxacin, ertapenem, and meropenem.

The main laboratory examinations performed during the admission are presented in Table 1.

The evolution was slowly favorable under antiviral treatment (remdesivir with a loading dose of 200 mg intravenously on the first day of admission, followed by 100 mg daily for the following 5 days), glucocorticoid medication (initially, the patient received dexamethasone 8 mg twice daily for 7 days followed by methylprednisolone 500 mg daily for 7 days), antibiotic therapy (initially, meropenem 4 g was administered daily for 10 days in the absence of the identified etiology in combination with linezolid 600 mg twice daily, and then ceftazidime-avibactam 2 g/0.5 g was administered three times daily for 7 days), voriconazole 200 mg twice daily for 14 days, anakinra 150 mg/mL with a loading dose of 2 ampoules daily for three days, followed by 1 ampoule daily for the following 7 days, with salbutamol inhaler 100 mcg per dose, corticosteroid inhaler budesonide/formoterol 160 mcg/4.5 mcg, and a slow reduction in oxygen requirement. The patient continued oxygen therapy at home after discharge with a flow rate of 5 L/minute; the patient was discharged after 28 days after his admission into the hospital. The patient’s evolution is summarized in Figure 3.

## 3. Discussion

*Lelliottia amnigena* is included in phylum *Pseudomonadota*, class *Gammaproteobacteria*, family *Enterobacteriaceae*, genus *Lelliotia* together with *L. nimipressuralis* (initially known as *Erwinia nimipressuralis* and then reclassified as *Enterobacter nimipressuralis*), and a new species *L. jeotgali*, which is associated with traditional Korean fermented clam [10]. Virulence factors are represented by pectinase, protease, pectin lyase, and cellulase [11].

*L. amnigena*, as previously mentioned, was reported in a few case reports of infections in humans, e.g., unilateral posttraumatic endophthalmitis in an immunocompetent patient [4], which was a case of urinary tract infection in a patient immunosuppressed known to have prostatic and colorectal adenocarcinoma, with obesity, nephrolithiasis, and nephrostomy [5]; a case of pyonephrosis [6], involving an infection in a heart transplant recipient [12]; and sepsis in a case report from Ethiopia [7].

The isolation of *L. amnigena* from sputum, blood, or stool is not always considered pathogenic and can be a matter of colonization [13]. Therefore, the diagnosis must be supported by clinical, paraclinical elements (imaging and laboratory studies), and bacteriological examination of samples. In our reported case, the patient presented with a febrile croup, hemoptysis (associated with an increasing rate of flow of the high-flow nasal oxygen therapy (a flow rate of 60 L/minute)), exacerbation of the acute respiratory failure (increasing CT scan severity score), increased lung lesions, and exacerbation of the inflammatory syndrome. Bacteriological examinations from the sputum at the time of admission and during evolution were negative. During the third harvesting of sputum samples, a sample that was derived from the site of an active infection based on the data from the microscopic examination of the sputum, *P. putida* was isolated along with *L. amnigena*, both of which are strains of low-virulence species and maintain susceptibility to antibiotics. In the context of an immunosuppressed patient with previous pulmonary surgery for actinomycosis, chronic obstructive pulmonary disease, and bronchiectasis, all these conditions are favorable for biofilm formation.

Biofilm-related infections (BRI) can take on many forms, ranging from catheter-associated urinary tract infections (which are still the most common type of BRI) to central line-associated bloodstream infections, cystic fibrosis, fracture-related infections, BRI associated with the use of fixed braces, and periprosthetic joint infections [14]. Although there is still no universally agreed upon definition of a biofilm, biofilms are defined as “a coherent cluster of bacterial cells imbedded in a biopolymer matrix, which, compared with planktonic cells, shows increased tolerance to antimicrobials and resists the antimicrobial properties of the host defence” [15]. *P. putida* is a Gram-negative bacteria belonging to the genus *Pseudomonas*, which is known to be associated with plant growth. *P. putida* strains are frequently isolated from the rhizosphere of plants and are known for their environmental involvement (capacity to degrade pollutants). It has previously been reported that *P. putida* has the ability to form bacterial biofilms and cell-to-cell communication systems [16]. C-di-GMP is a main regulator factor of biofilm formation for *P. putida* strains, which is also involved in the regulator’s mechanisms of the LapA adhesion protein on the cell surface. LapA adhesion protein seems to be one of the most important biofilm matrix structures for biofilm formation by *P. putida*. AHL-based LuxIR-type cell-to-cell communication system signaling is functional in *P. putida* biofilms, although the true expression of a quorum sensing mechanism depends on the strain [17].

In terms of the antibiotic susceptibility test results, *L. amnigena* is considered to be naturally resistant to first-generation cephalosporins and cefoxitin. The resistance to ampicillin is reported in over 83% of isolates strains and to amoxicillin/clavulanic acid in over 33% of isolates [18]. Other authors also describe resistance to doxycycline, gentamicin, and beta-lactam/beta-lactamase inhibitors [19], but also decreased sensitivity to cefixime, cefpodoxime, and ceftibuten [20]. In our case, we found no resistance to other antibiotics except the one described as natural resistance in the isolated strain identified by us.

## 4. Conclusions

*L. amnigena* is a pathogenic microorganism that exhibits relatively weak virulence. It is infrequently encountered in human infections, mostly with compromised immune systems, and has been isolated from bodily fluids such as sputum, blood, and urine. However, it should be noted that *L. amnigena* is not always the primary causative agent of the infection but rather a potential colonizer. The simultaneous presence of another pathogenic bacteria with low virulence can increase *L.amnigena* virulence, causing a polymicrobial infection in circumstances similar to our case report, involving an infection related to biofilm. Identifying specific pathogens via routine methods in microbiology laboratories can pose a challenge. Moreover, the susceptibility to third-generation cephalosporins may be compromised when dealing with a limited number of isolates. *L. amnigena* remains a pathogen rarely isolated in human pathology, even if there are concerns among some readers if *L. amnigena* is a colonizer or an actual infection in this clinical setting. Still, we should pay more attention, especially to immunosuppressed patients, where it can be responsible for a dire clinical picture.

## Figures and Tables

**Figure 1 microorganisms-11-02143-f001:**
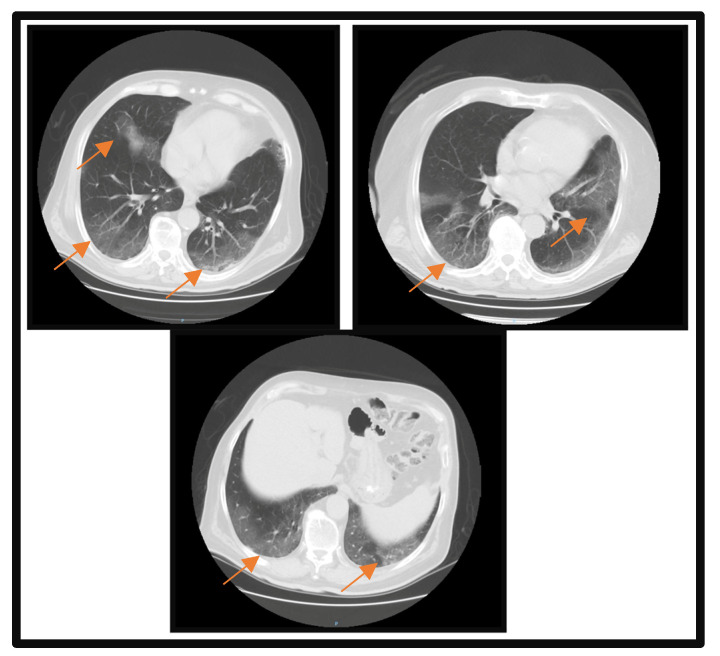
Pulmonary CT scan at the time of admission. Orange arrows highlight the lesion areas of condensation in the matt glass aspect.

**Figure 2 microorganisms-11-02143-f002:**
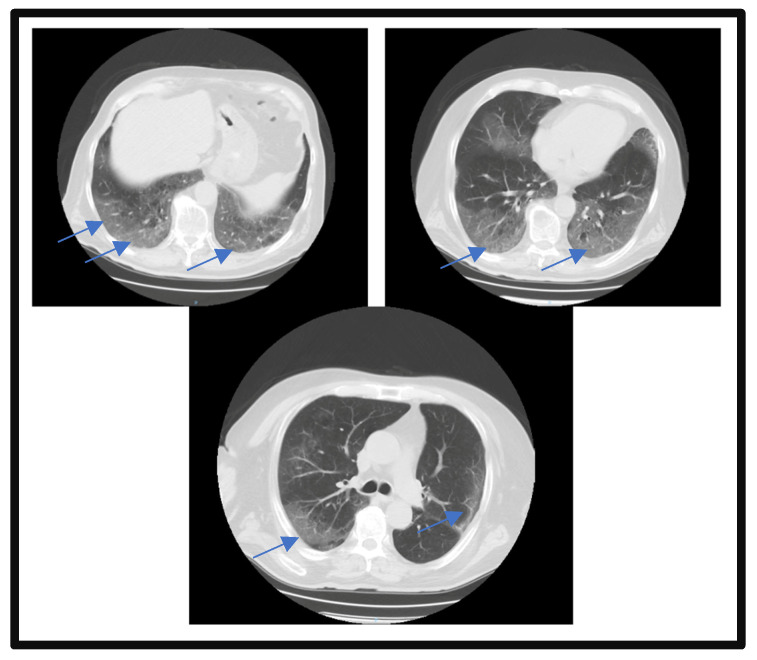
Pulmonary CT scan on the 3rd day of admission. Blue arrows highlight the lesion areas of condensation in the matt glass aspect.

**Figure 3 microorganisms-11-02143-f003:**
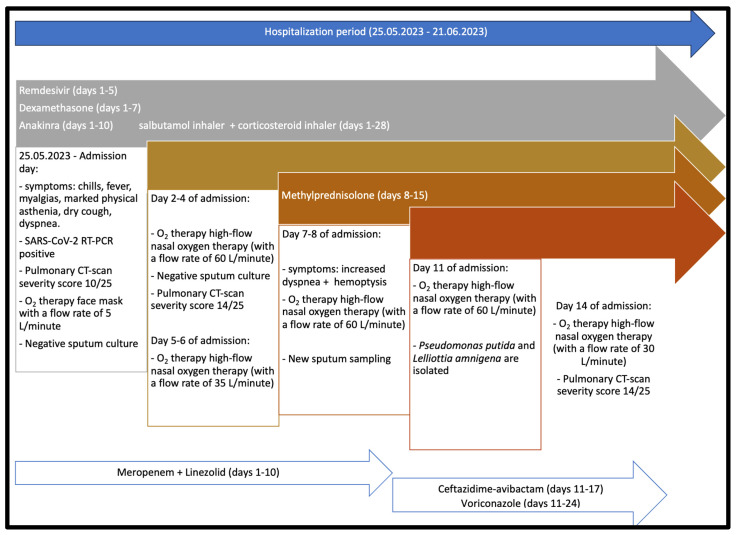
Case report flow diagram.

**Table 1 microorganisms-11-02143-t001:** Laboratory examinations during hospitalization.

Date	Parameter	Values	Reference Value
On admission	C-Reactive Protein	147.22 mg/L	0–5 mg/L
	Serum amylase	27 U/L	28–100 U/L
	eGFR	95.75 mL/min/1.73 m^2^	
	Aspartate aminotransferase	47 U/L	11–34 U/L
	WBCs Differential blood count: Neutrophils Lymphocytes Monocytes Basophils Eosinophils	10.76 × 10^3^/µL 8.75 × 10^3^/µL 1.24 × 10^3^/µL 0.74 × 10^3^/µL 0.02 × 10^3^/µL 0.01 × 10^3^/µL	4–10 × 10^3^/µL 2–7.5 × 10^3^/µL 1.5–4 × 10^3^/µL 0.2–1 ×10^3^/µL 0–0.2 × 10^3^/µL 0–0.7 × 10^3^/µl
	Blood glucose	131 mg/dL	80–115 mg/dL
	Ratio of neutrophils to lymphocytes	7.056	
30 May 2023 (6th day of hospitalization)	C-Reactive Protein	14.28 mg/L	0–5 mg/L
	Fibrinogen	439.8 mg/dL	170–420 mg/dL
	eGFR	99.28 mL/min/1.73 m^2^	
	Blood glucose	179 mg/dL	80–115 mg/dL
	ESR	21 mm/h	0–15 mm/h
	Ratio of neutrophils to lymphocytes	8.976	
	WBCs Haemoglobin Hematocrit Thrombocytes	8.83 × 10^3^/µL 13.9 g/dL 40.6% 186 × 10^3^/µl	4–10 × 10^3^/µL 13–17 g/dL 40–50% 150–400 × 10^3^/µL
3 June 2023 (9th day of hospitalization)	C-Reactive Protein	3.27 mg/L	0–5 mg/L
	Fibrinogen	285.1 mg/dL	170–420 mg/dL
	eGFR	96.88 mL/min/1.73 m^2^	
	Blood glucose	189 mg/dL	80–115 mg/dL
	D-dimers	1217.53 ng/mL	45–499 ng/mL
	Fibrin monomers	Positive	Negative
	Ratio of neutrophils to lymphocytes	14.704	
	WBCs Haemoglobin Hematocrit Thrombocytes	13.38 × 10^3^/µL 15.2 g/dL 44.4% 304 × 10^3^/µL	4–10 × 10^3^/µL 13–17 g/dL 40–50% 150–400 × 10^3^/µL
	WBCs Differential blood count: Neutrophils Lymphocytes Monocytes Basophils Eosinophils	11.91 × 10^3^/µL 0.81 × 10^3^/µL 0.65 × 10^3^/µL 0.01 × 10^3^/µL 0.00 × 10^3^/µL	2–7.5 × 10^3^/µL 1.5–4 × 10^3^/µL 0.2–1 × 10^3^/µL 0–0.2 × 10^3^/µL 0–0.7 × 10^3^/µL

ESR (erythrocyte sedimentation rate).

## Data Availability

All data generated or analyzed during this study are included in this published article.

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
