# Peer review of "Lelliottia amnigena and Pseudomonas putida Coinfection Associated with a Critical SARS-CoV-2 Infection: A Case Report"

_microorganisms, 2023, doi:10.3390/microorganisms11092143_

Round 1
Reviewer 1 Report
The main advantages of this case report
· The article had a good level of English.
· The authors have no conflicts of interest.
· The article presents an interesting topic.
· The article has multiple bibliographic sources
· The article is well-defined.
· The work suits the journal scope.
· The article is of elevated quality.
The main disadvantages of this case report:
· Center text above image
· How many vaccines did he have?
· If the patient was previously infected with covid?
· Complete reference values ​​on all days
· What medications was the patient taking?
· Specify the doses for each drug.
· The conclusion still needs to be elaborated (is to short)
· How many days was the patient hospitalized? His evolution, maybe another CT scan
· The article is limited to several people.
Author Response
Sibiu, 6.08.2023
To
the Editors of Healthcare®
Dear Editor-in-Chief,
Dear Editor,
Dear Reviewer,
Thank you for reviewing our manuscript. Please find attached a revised version of our manuscript, “Lelliottia amnigena and Pseudomonas putida coinfection associated with a critical SARS-CoV-2 infection. A case report.”.
Thank you for taking the time to review our revised manuscript. Your insightful comments, along with those of our other reviewers, were incredibly helpful in improving the quality of our paper.
We are pleased to inform you that we have carefully considered each of your comments and have made the necessary revisions to the manuscript. We have attached a point-by-point response to each of your comments to provide you with a better understanding of how we have addressed your concerns.
We greatly appreciate your time and effort in reviewing our manuscript and providing invaluable feedback. Your contributions have undoubtedly strengthened our work, and we are grateful for your support.
Reviewer Comments:
Reviewer 1
The main advantages of this case report
- The article had a good level of English.
- The authors have no conflicts of interest.
- The article presents an interesting topic.
- The article has multiple bibliographic sources
- The article is well-defined.
- The work suits the journal scope.
- The article is of elevated quality.
Answer: We greatly appreciate your time and effort in reviewing our manuscript and providing invaluable feedback. Your contributions have undoubtedly made our work more substantial, and we are grateful for your support.
The main disadvantages of this case report:
- Center text above image
Answer: Thank you for the suggestion. We performed the necessary changes.
- How many vaccines did he have?
A: Thank you for the question. The immunization status of the patient was included in the case report.
- If the patient was previously infected with covid?
A: We thank the distinguished Reviewer for highlighting this aspect. The patient was not previously diagnosed with COVID-19 or other new respiratory symptoms.
- Complete reference values ​​on all days
A: Thank you for the suggestion. We performed the necessary changes.
- What medications was the patient taking?
- Specify the doses for each drug.
A: We thank the distinguished Reviewer for highlighting this aspect. We included a detailed list of the medication that the patient received during the hospitalization period.
- The conclusion still needs to be elaborated (is to short)
A: thank you for this suggestion. The conclusion section of the manuscript was elaborated.
- How many days was the patient hospitalized? His evolution, maybe another CT scan
A: We thank the distinguished Reviewer for highlighting this aspect. The patient was discharged after 28 days after his admission into the hospital. Three CT scans were performed during the hospitalization period, the last on the 14-day admission.
- The article is limited to several people.
A: We thank the distinguished Reviewer for highlighting this aspect. Our case report is valuable and will improve the knowledge of this rare pathogen that also causes human infections.
We hope that the revised form of the manuscript and our accompanying responses will be sufficient to make our manuscript suitable and accepted for publication in Microorganisms®. We shall look forward to hearing from you at your earliest convenience.
With our best regards,
Sincerely yours,
Victoria Birlutiu, Prof. Habil. M.D. Ph.D
Rares Mircea Birlutiu, M.D. Ph.D.
Reviewer 2 Report
This is a case report of patient admitted with COVID, and who had Lelliottia amnigena and Pseudomonas putida infection during hospital admission. It seems the author’s point is the rarity of the organism, Lelliottia amnigena, in a clinical setting. As it was isolated from sputum which is not a normally sterile culture, the isolated organisms may not necessarily represent pathogenic. It is especially true in the hospitalized patient who is on broad-spectrum antibiotics. The authors need to describe why there is a good reason to believe that organism is a pathogen. (For example, the organism was isolated from good quality sputum or pathogen was isolated multiple times, etc) If they are unable to provide the information, the case should be “Lelliottia aminigena and Pseudomonas putida isolated from sputum sample in a patient with critical SARS-CoV2 infection”.
My specific comments are as below,
1. Case report, 2nd paragraph (Lines 61-70) – “A severity score of 10 out of 25 was reported” – what is the severity score? It should be specified. And this paragraph includes a lot of radiographic findings unrelated to the case report. It should be summarized to include only relevant findings.
2. Case report – the time course is very difficult to follow. I would recommend creating a figure including time course with timing of events and microbiology tests, antimicrobials used, oxygen requirement, and so on. That will greatly help readers understand the clinical course.
3. As I mentioned above, isolation from sputum does not mean the organism is pathogenic. The authors should report how they collected the sputum sample and the quality of sputum sample to justify those organisms were from lower respiratory tract rather than oral contamination.
4. Hemoptysis was the reason for additional sputum culture. Hemoptysis can be caused by many reasons including infection and non-infection. Was that episode related to increased oxygen requirement or worsening radiographic findings? The figure of clinical course would help clarify that.
5. Table 1, not all labs need to be reported. Should choose ones relevant to the case. Follow-up labs should be tagged as day___ rather than actual date.
6. Line 102 – corticotherapy? Does it mean corticosteroid? It is an uncommon term and I suggest to change to corticosteroid.
7. Line 104 – could authors explain why broad-spectrum antibiotics and voriconazole were used in the absence of organisms at the beginning? It is unfortunately a common practice but would be good to state if there was a convincing reason to do so.
8. Discussion – Line 119-120. It is not relevant to this case and can be removed.
9. Lines 121-123 “Isolation of ….. Paraclinical elements” is a true statement. But the authors included symptoms and findings of whole clinical course to justify in the following sentences. It should be focused on the event of hemoptysis while on treatment. Please revise.
10. Lines 132-149. This paragraph discuss about biofilm formation of Pseudomonas putida. It reads very isolated from other parts of discussion for L. amnigena. I recommend removing that part.
Author Response
Sibiu, 6.08.2023
To
the Editors of Microorganisms®
Dear Editor-in-Chief,
Dear Editor,
Dear Reviewer,
Thank you for reviewing our manuscript. Please find attached a revised version of our manuscript, “Lelliottia amnigena and Pseudomonas putida coinfection associated with a critical SARS-CoV-2 infection. A case report.”.
Thank you for taking the time to review our revised manuscript. Your insightful comments, along with those of our other reviewers, were incredibly helpful in improving the quality of our paper.
We are pleased to inform you that we have carefully considered each of your comments and have made the necessary revisions to the manuscript. We have attached a point-by-point response to each of your comments to provide you with a better understanding of how we have addressed your concerns.
We greatly appreciate your time and effort in reviewing our manuscript and providing invaluable feedback. Your contributions have undoubtedly strengthened our work, and we are grateful for your support.
Reviewer Comments:
Reviewer 2
This is a case report of patient admitted with COVID, and who had Lelliottia amnigena and Pseudomonas putida infection during hospital admission. It seems the author’s point is the rarity of the organism, Lelliottia amnigena, in a clinical setting. As it was isolated from sputum which is not a normally sterile culture, the isolated organisms may not necessarily represent pathogenic. It is especially true in the hospitalized patient who is on broad-spectrum antibiotics. The authors need to describe why there is a good reason to believe that organism is a pathogen. (For example, the organism was isolated from good quality sputum or pathogen was isolated multiple times, etc) If they are unable to provide the information, the case should be “Lelliottia aminigena and Pseudomonas putida isolated from sputum sample in a patient with critical SARS-CoV2 infection”.
Answer: We greatly appreciate your time and effort in reviewing our manuscript and providing invaluable feedback. Your contributions have undoubtedly made our work more substantial, and we are grateful for your support. We hope that with the newly included information that is present in lines 106-125 there is no doubt that two isolated bacteria are pathogens and not contaminators.
My specific comments are as below,
- Case report, 2nd paragraph (Lines 61-70) – “A severity score of 10 out of 25 was reported” – what is the severity score? It should be specified. And this paragraph includes a lot of radiographic findings unrelated to the case report. It should be summarized to include only relevant findings.
A: We thank the distinguished Reviewer for highlighting this aspect. The score was proposed by Francone M, please see the ref. Francone M, Iafrate F, Masci GM, et al. Chest CT score in COVID-19 patients: correlation with disease severity and short-term prognosis. Eur Radiol. 2020;30(12):6808-6817. doi:10.1007/s00330-020-07033-y. The score is reported based on the authors and not summarized so a complete picture of the lung involvement can be assessed.
- Case report – the time course is very difficult to follow. I would recommend creating a figure including time course with timing of events and microbiology tests, antimicrobials used, oxygen requirement, and so on. That will greatly help readers understand the clinical course.
A: A case report flow diagram was included into the manuscript as per the distinguished Reviewer suggestion.
- As I mentioned above, isolation from sputum does not mean the organism is pathogenic. The authors should report how they collected the sputum sample and the quality of sputum sample to justify those organisms were from lower respiratory tract rather than oral contamination.
A: As per the distinguished Reviewer question, the following information was included into the manuscript. We hope that with the newly included information that is present in lines 85-99 there is no doubt that two isolated bacteria are pathogens and not contaminators.
- Hemoptysis was the reason for additional sputum culture. Hemoptysis can be caused by many reasons including infection and non-infection. Was that episode related to increased oxygen requirement or worsening radiographic findings? The figure of clinical course would help clarify that.
A: A case report flow diagram was included into the manuscript as per the distinguished Reviewer suggestion. The hemoptysis was associated with an increase of oxygen requirement.
- Table 1, not all labs need to be reported. Should choose ones relevant to the case. Follow-up labs should be tagged as day___ rather than actual date.
A: We thank the distinguished Reviewer for highlighting this aspect. We also included the day of hospitalization, we will keep all reported information as per the suggestion of the other Reviewers.
- Line 102 – corticotherapy? Does it mean corticosteroid? It is an uncommon term and I suggest to change to corticosteroid.
A: We thank the distinguished Reviewer for highlighting this aspect, the term was changed to glucocorticoid medication.
- Line 104 – could authors explain why broad-spectrum antibiotics and voriconazole were used in the absence of organisms at the beginning? It is unfortunately a common practice but would be good to state if there was a convincing reason to do so.
A: The attending physician decided on the broad-spectrum antibiotic therapy given the severity of the case.
- Discussion – Line 119-120. It is not relevant to this case and can be removed.
A: We thank the distinguished Reviewer for highlighting this aspect, it might not be relevant, but due to the rarity of the isolated species, there are 3 species, we think that it can also be reported.
- Lines 121-123 “Isolation of ….. Paraclinical elements” is a true statement. But the authors included symptoms and findings of whole clinical course to justify in the following sentences. It should be focused on the event of hemoptysis while on treatment. Please revise.
A: Thank you for bringing up this aspect. We included more information in the paragraph.
- Lines 132-149. This paragraph discuss about biofilm formation of Pseudomonas putida. It reads very isolated from other parts of discussion for L. amnigena. I recommend removing that part.
A: Thank you for bringing up this aspect. While it may not seem immediately relevant, we appreciate your input on the matter. Given the rarity of the isolated species in question, it may be worth including in the report. Thank you again for your valuable feedback. The information on P. putida biofilm formation underlines the possibility of polymicrobial biofilms to increase some strains' virulence.
We hope that the revised form of the manuscript and our accompanying responses will be sufficient to make our manuscript suitable and accepted for publication in Microorganisms®. We shall look forward to hearing from you at your earliest convenience.
With our best regards,
Sincerely yours,
Victoria Birlutiu, Prof. Habil. M.D. Ph.D
Rares Mircea Birlutiu, M.D. Ph.D.
Reviewer 3 Report
The paper titled Lelliottia amnigena and Pseudomonas putida co-infection associated with critical SARS-CoV-2 infection. Case report requires completion.
In my opinion, the indications for microbiological examination, the method of collecting clinical material and the details of the examination and interpretation should be described in detail. Only this will allow you to determine whether they are true pathogens, whether their presence in the sample is related to the colonization of the patient.
Please see the attached file for my comments and suggestions.

This work is written quite accurately in English. Minor editing of English language required.
Author Response
Sibiu, 6.08.2023
To
the Editors of Microorganisms®
Dear Editor-in-Chief,
Dear Editor,
Dear Reviewer,
Thank you for reviewing our manuscript. Please find attached a revised version of our manuscript, “Lelliottia amnigena and Pseudomonas putida coinfection associated with a critical SARS-CoV-2 infection. A case report.”.
Thank you for taking the time to review our revised manuscript. Your insightful comments, along with those of our other reviewers, were incredibly helpful in improving the quality of our paper.
We are pleased to inform you that we have carefully considered each of your comments and have made the necessary revisions to the manuscript. We have attached a point-by-point response to each of your comments to provide you with a better understanding of how we have addressed your concerns.
We greatly appreciate your time and effort in reviewing our manuscript and providing invaluable feedback. Your contributions have undoubtedly strengthened our work, and we are grateful for your support.
Reviewer Comments:
Reviewer 3
The paper titled Lelliottia amnigena and Pseudomonas putida co-infection associated with critical SARS-CoV-2 infection. Case report requires completion. In my opinion, the indications for microbiological examination, the method of collecting clinical material and the details of the examination and interpretation should be described in detail. Only this will allow you to determine whether they are true pathogens, whether their presence in the sample is related to the colonization of the patient.
Answer: We greatly appreciate your time and effort in reviewing our manuscript and providing invaluable feedback. Your contributions have undoubtedly made our work more substantial, and we are grateful for your support. We hope that with the newly included information that is present in lines 106-125 there is no doubt that two isolated bacteria are pathogens and not contaminators.
Below are the main comments
Minor
- Line 45, Introduction section; The reference number is missing.
A: We thank the distinguished Reviewer for highlighting this aspect. The reference number was added.
- Line 47; ,,germs,, change to bacteria or microorganisms
A: We thank the distinguished Reviewer for highlighting this aspect. The term was changed as recommended.
- Figure no. 1 and Figure no. 2 – change to Figure 1 and Figure 2, Figures have shown CT images without a description can be incomprehensible for the Microorganisms reader. Arrows indicating e.g. consolidations or other changes in the lung parenchyma can be helpful.
A: The title of the figures was changed as recommended. We included some arrows in the CT scan sections that highlight the lesion areas of condensation in matt glass aspect.
Major
- Because the study deals with the isolation of relatively rarely occurring bacterial species from a patient with SARS-CoV-2, efforts should be made to provide a more detailed description of the microbiological investigations. Their interpretation is crucial in determining the etiology of infection.
The authors should provide more detailed information in their study regarding the microbiological investigations. They should mention:
- The biological material from which the bacteria were isolated (the number of samples collected and how many of them yielded bacterial growth).
A: We thank the distinguished Reviewer for highlighting this aspect. We hope that with the newly included information that is present in lines 106-125 there is no doubt that two isolated bacteria are pathogens and not contaminators. Two sample was harvested each time in the morning (both samples were positive).
- The day of hospitalization when the sample was collected for microbiological testing.
A: The day of hospitalization was included into the text, also we added a flow diagram of the case.
- Whether the microbiological testing was quantitative (providing the bacterial count) or qualitative (identifying the presence/absence of specific bacteria).
A: Unfortunately, in the results the ID specialists receive from the lab the CFU are reported only for sonication fluid cultures, we do not have the number of CFU for this case.
- Whether the patient was on antibiotics before the microbiological testing was requested.
A: We thank the distinguished Reviewer for highlighting this aspect. Before the first sampling the patient was not under antibiotic therapy, nor in the last 3 months prior to his admission into the hospital.
- How long did the microbiological examination last.
A: The samples were positive after 3 days
In microbiological studies, the sampling method is crucial, and it can significantly impact the interpretation of the results, especially when dealing with bacteria commonly found in the environment (soil, water), or food. These bacteria may contaminate the sample, which is unrelated to the actual infection. Providing this additional information will help ensure a more accurate and reliable interpretation of the study findings. Only in the Discussion section, the authors described that microbiological tests of sputum at the time of admission and later were negative (at that time there were symptoms of infection and significantly elevated CRP)
A: We thank the distinguished Reviewer for highlighting this aspect. The following information was included also in the case report (“During the first week of hospitalization two bacteriological examination of the specimens obtained from the lower respiratory tract (sputum) were performed but all cultures were negative.”). please see line 109-125
- Line 89; and the references of the EUSACT recommendations. Complete the Reference section.
A: We thank the distinguished Reviewer for highlighting this aspect. We reassessd the lab report for the information regarding the EUCAST AST, and the results were assesste based on the CLSI breakpoints, we corrected the information in the manuscript. Minimum inhibitory concentrations were assessed according to the Clinical and Laboratory Standards Institute (CLSI) MIC breakpoints (CLSI M100 32th edition) ref.[9]. Thank you for pointing this fact.
- It is suggested that Lelliottia amnigena is naturally resistant to second- and third-generation cephalosporins. The authors wrote that the isolated strain of L. amnigena is sensitive to ceftriaxone and ceftazidime. Did the authors verify that the strain was correctly identified? Was the identification repeated? Which Vitek system library was used, and what was the identification score? By what method the antibiogram was performed?
A: L. amnigena is considered to be naturally resistant to first-generation cephalosporins and cephamycin (cefoxitine), not to 3rd generation, ref 21 suggest decrease of susceptibility but not natural resistance. The strain was correctly identified. Two samples were haversted and bot were positive for the isolated strains. Unfortunately were don’t have information as ID specialist regarding the identification score. To our knowledge Vitek 2 System Version 9.01 is installed, unfortunately we don’t know it version 9.02 was installed. The ID used card is GN ref 21341. The antibiogram was performed using the automated AST system from Vitek 2. Thank you for pointing this facts.
- On day 6, the CRP concentration was already significantly reduced, also the WBC count. What were the grounds for suspicion of bacterial infection and indications for microbiological examination.
A: We thank the distinguished Reviewer for highlighting this aspect. A case report flow diagram was included into the manuscript. The hemoptysis was on day 7/8 and associated with an increase of oxygen requirement. The patients was already under antiviral treatment and broad-spectrum antibiotics. Thank you for pointing this fact.
- Whether the information on which day the empirical antibiotic therapy was started is known.
A: On the first day of admission, details regarding the therapy have been included in the manuscript. Thank you for pointing this fact.
- In discussion section, a paragraph regarding L. nimipressuralis is unlikely to relate to the subject of the work. The text about biofilm in the Discussion is also not directly related to the case description.
A: Thank you for bringing up this aspect. While it may not seem immediately relevant, we appreciate your input on the matter. Given the rarity of the isolated species in question, it may be worth including in the report. The information on P. putida biofilm formation underlines the possibility of polymicrobial biofilms to increase some strains' virulence. Thank you again for your valuable feedback.
We hope that the revised form of the manuscript and our accompanying responses will be sufficient to make our manuscript suitable and accepted for publication in Microorganisms®. We shall look forward to hearing from you at your earliest convenience.
With our best regards,
Sincerely yours,
Victoria Birlutiu, Prof. Habil. M.D. Ph.D
Rares Mircea Birlutiu, M.D. Ph.D.
Round 2
Reviewer 1 Report
The paper can be accepted without any further changes.
Author Response
We greatly appreciate your time and effort in reviewing our manuscript and providing invaluable feedback. Your contributions have undoubtedly made our work more substantial, and we are grateful for your support during the revision step.Reviewer 2 Report
In this revised manuscript, the authors provided revisions according to suggestions from me and other reviewers, which improved the manuscript. I still have a couple of comments.
1. The authors provided an evidence the quality of sputum sample was good suggesting the bacteria were from lower respiratory tract, not oropharyngeal space. The authors claimed undoubtedly they are pathogens, but it is not true. It is possible the hemoptysis was from non-infectious cause and those bacteria were just by-standers. (It is especially true when the bacteria was not seen on Gram stain, so the number of bacteria was very low) The only situation they can definitively say those environmental bacteria are pathogens is when those are isolated from usually sterile site such as blood culture. For this report, I suggest adding a sentence in limitation section that those isolates may not represent true pathogens even if the quality of sputum sample was good.
2. I still think the discussion about L. nimipressuralis (Lines 164-165) and Pseudomonas putida (Lines 181-198) are irrelevant and should be removed. It was pointed out by another reviewer (for L. nimipressuralis section). The rarity is not an appropriate reason to include in the discussion.
Author Response
Dear Reviewer,
Thank you for taking the time to reassess our manuscript. We highly appreciate your comments and would like to express our respect for them. We have carefully considered your suggestion to exclude line 164-165 from the new version of the manuscript, which states that “3. Discussion Lelliottia amnigena is included in phylum Pseudomonadota, class Gammaproteobacteria”. If you recommend removing this line, we kindly request an official request by the Editor in Chief of the journal.
However, we do agree with the Reviewer's suggestion to remove line 176-177, based on the comment provided.
We would not also be addressing comment no.1 and no.2 (the second suggestion).
We would like to assure you that we will accept the final decision of the Editor in Chief of the Journal regarding this submission, regardless of the outcome.
Thank you again for your time and effort in reviewing our manuscript.
Sincerely,
The Authors
Reviewer 3 Report
Most of my questions and suggestions have taken into account. Further clarification is needed. regarding the recommendation for a microbiological examination on the 7th day. As evident from the case description, symptoms and test results do not indicate a bacterial infection.
This clarification is necessary for the interpretation of the microbiological test results. The isolation of bacteria does not necessarily indicate an infection and does not require antibiotic treatment; it is possible that the patient (treated with a carbapenem, i.e. a broad-spectrum antibiotic) was colonized during hospitalization? Such critical analysis should be described in the discussion section.
Author Response
We thank the distinguished Reviewer for highlighting this aspect. We have answered all of the Reviewer's questions and considered almost all suggestions. We highly appreciate your comments and would like to express our respect for them; we do not believe that clarifications are needed regarding the microbiological reexamination on the 7th day of hospitalization. Isolation of a bacteria in the clinical setting of a patient that has increased requirements of oxygen therapy and all the other manifestations suggests that the isolated strain is pathogenic. Regarding the Reviewer's concern about colonization, the isolated strain was never isolated previously in the hospital. If the Reviewer knows about cases of HAI with this bacteria, we suggest he also share his knowledge with us. We would also like to mention the isolate’s rarity, considering the literature review. We updated the conclusion of the manuscript.
We have no further comments on this aspect.
We would like to assure you that we will accept the final decision of the Editor in Chief of the Journal regarding this submission, regardless of the outcome.
Thank you again for your time and effort in reviewing our manuscript.
Sincerely,
The Authors